# OpenReview forum: "Unveiling the Bias Impact on Symmetric Moral Consistency of Large Language Models"
_NeurIPS.cc/2024/Conference — NeurIPS 2024 poster_

### Official Review · Reviewer_SWHZ · 2024-07-05

**Soundness:** 2
**Presentation:** 3
**Contribution:** 2
**Rating:** 5
**Confidence:** 4

**Summary:**

This paper proposes a framework called tSMC that uses KL divergence to measure the consistency of 12 LLMs on a moral question dataset. They find that the LLMs are more consistent on low-ambiguity questions, and less so on high-ambiguity questions. They also find that LLMs are susceptible to both position and selection bias, which their proposed measure corrects for.

**Strengths:**

I liked the investigation into confidence distributions instead of only prompt-based approaches. Bold sentences throughout helped to make clear the important takeaways.

**Weaknesses:**

The central premise of this paper I find to be a bit confusing: why is moral consistency desired? Why is higher consistency framed as “better” (L185) and good throughout? From reading Figure 1, I’m not convinced any answer is right (e.g., what if the son is 30 years old and in a state where smoking weed is legal?). What if the answers are consistently more immoral, certainly the “right answer” is also relevant here rather than only consistency in isolation? Also, what about this framework is specific to moral questions, if any? Could it be generalized to consistency measures in general? All of these questions were on my mind as I read the paper.

Other weaknesses:
- L47-51: I would like a lot more explanation of this metric if it is being introduced here. What is the “standard method”? As well as some intuition to understand Table 1 which is presented here. If “Avg” is a measure of consistency, how should I be thinking about “Std,” which is sometimes also treated as a measure of consistency. Is it instead to be treated as some sort of confidence measure in the consistency?
- Why is “A(B)” and “C(D)” used? Is this common? I feel like I normally would just say A vs B for a two-option multiple choice.
- L84-85: “commonly encountered in practical applications” —> what is the evidence or citation for this?
- L134: “context swap” and “full swap” have not been defined yet but are just used here
- L156: I find the naming of this as “true moral consistency” to be a bit of an overclaim, if anything it is “adjusted moral consistency” since while it may account for position and selection bias, it doesn’t account for other points of bias, e.g., prompt word choice.
- Table 2: if [9] is the main point of comparison, this should really be explained more as well as the details of what exactly is different about your method. Also, just because your measure is different from their method, how do we assess which is a “better” measure?
- I would appreciate consistency (pun intended), in how you refer to the same phenomenon as either “inconsistency” or “bias” (e.g., L219-221)
- L231-236: is this also true of Llama-2-7b? This feels like a potential overclaim
- I loved the use of LLM confidence scores in 2.4, but I what I really wanted to see here was a direct comparison of the confidence scores to the consistency scores. Are they trying to measure the same thing? Do they exhibit the same thing?
In this particular paper, I think the related work might have made more sense earlier in the paper to set context, as well as explain more about [9].

**Questions:**

Beyond the questions I listed above in weaknesses (as they prevented me from fully buying into the paper), I am also wondering whether your method would scale to questions with more than two MC answer choices?

**Limitations:**

Addressed

---

> ### Author Rebuttal · Authors · 2024-08-06
>
> Thank you for the constructive feedback.
>
> > **[W0.1] Why value moral consistency? Why is higher consistency better? What if choices are immoral?**
>
> Moral consistency is desired in AI systems to ensure reliability and predictability in moral reasoning ability. Better refers to the model's consistency in reasoning, not moral superiority. We will revise it to make this distinction clearer in our manuscript.
>
> In our work, we assess both low-ambiguity and high-ambiguity scenarios. In high-ambiguity scenarios, both choices present obvious ethical flaws [1], corresponding to the consistently immoral decisions and the moral consistency metric can reflect LLMs' actual moral beliefs. In these circumstances, LLMs' consistent preferences become particularly valuable for further in-depth research into developing moral reasoning capabilities in LLMs.
>
> > **[W0.2] Generalization**
>
> **Table 1** in the **rebuttal PDF** shows GPT-3.5-Turbo's results on MMLU high_school_us_history [2], demonstrating our framework's broader applicability. CS, OS, and FS experiments reveal significant variations in consistency and accuracy across general knowledge tasks, proving our approach can detect biases and inconsistencies in LLM responses beyond ethical scenarios.
>
> > **[W1] Metric**
>
> Assume that $e$ is one prompt format and $e_N$ is a perturbed version (e.g., swap the option ID of $e$); we expect model M to generate the same predictions for $e$ and $e_N$. The consistency score $T$ on dataset $\mathcal{E}$ can be observed as follows:
> $$
> \begin{align}
> T = \frac{1}{|\mathcal{E}|} \sum_{e \in \mathcal{E}} \mathbb{I}(M(e) = M(e_N)).
> \end{align}
> $$
> The "standard method" refers to the method used by [9],  which adopts the above metric to evaluate symmetric consistency. While their method assesses only one type of perturbation (Context Swap), we have identified potential influences of biases on the results. Consequently, we have introduced two additional experimental settings, Option Swap and Full Swap, to validate these findings.
>
> "Std" in our analysis serves as a measure of consistency invariance across different experimental settings. A lower Std indicates that the model maintains similar consistency levels regardless of the setting.
>
> > **[W2] A(C)**
>
> Sorry for any confusion. We evaluate the model results separately for options AB and CD to assess whether LLMs prefer particular option identifiers (e.g., A). The results for options CD are presented in **Appendix B**. The notation A(C) in **Figure 1** only indicates that we conduct two distinct groups of experiments.
>
> > **[W3] Citation on L84-85**
>
> In practical applications, users formulate prompts in diverse ways, leading to structural variations such as reordering of options. These variations in prompt structure can result in different model responses [3]. We will incorporate appropriate citations to support this revised statement in our manuscript.
>
> > **[W4] L134**
>
> We have defined the concepts of Context Swap, Option Swap, and Full Swap in the caption of **Figure 2**, along with an intuitive demonstration of how these settings work in **Figure 2**. We will clarify this further in the revision.
>
> > **[W5] Account for other biases**
>
> While our framework addresses these specific biases (it is already non-trivial), we recognize the potential for extending it to encompass additional factors. In the revised paper, we will discuss possible extensions of our methodology to account for other types of biases, such as those arising from prompt word choice.
>
> > **[W6] Difference with [9]**
>
> As noted in [W1], our work differs significantly from [9] in several key technical aspects. Unlike [9], which focuses on symmetric consistency in Natural Language Inference tasks, our study evaluates the moral consistency of LLMs and finds that the apparent consistency is significantly affected by position and selection biases. Besides, we propose the framework tSMC to effectively quantify and mitigate these biases in the results, which we consider the "better" measure to achieve.
>
> > **[W7] Inconsistency or bias**
>
> Our experiments demonstrate that biases influence the apparent consistency scores of LLMs. We quantify these biases using Kullback-Leibler divergence as detailed in Equations 1 and 2. **Figure 3** illustrates the biases present in different models. Our tSMC method effectively mitigates these biases, as shown in **Figure 1** of the **rebuttal PDF**. This approach aims to present the model's moral consistency without the confounding effects of biases.
>
> > **[W8] Llama-2-7b**
>
> Llama-2-7b exhibits similar tendencies, albeit slightly less than 13b/70b. We recognize that the underlying causes for these observations are not definitively established. Considering the potential for multiple contributing factors (e.g., post-training), we will revise our discussion to reflect this complexity and the need for further investigation.
>
> > **[W9] Comparison; Related work**
>
> To address this, we conduct a direct comparison. Please refer to **Figure 2** of the **rebuttal PDF**. These results indicate that confidence and consistency scores measure differently. Confidence scores reflect the model's certainty in its responses while consistency scores indicate the model's ability to provide consistent judgments.
>
> We propose to move the related work to follow the introduction. We'll explain how [9] introduced symmetric consistency in NLI tasks, highlighting our extension to moral scenarios and additional biases consideration.
>
> > **[Q] Scale to more choices**
>
> Our work focuses on "symmetric" consistency that is non-trivial. The "symmetric" means the choice number should be two. We will explore the cases of more choices in the future.
>
> **References**
>
> [1] Scherrer et al. Evaluating the Moral Beliefs Encoded in LLMs. NeurIPS 2023.
>
> [2] Hendrycks et al. Measuring Massive Multitask Language Understanding. ICLR 2021.
>
> [3] Zhao et al. Calibrate Before Use: Improving Few-shot Performance of Language Models. ICML 2021.

---

> ### Author Response · Authors · 2024-08-13
>
> Dear Reviewer,
>
> Thank you very much for your time and efforts in reviewing our paper. We truly value your insights and have diligently addressed them in our rebuttal.
>
> As we greatly esteem your expertise, we kindly request that you take a moment to review our responses. We're eager to address any additional feedback, questions, or clarifications you might have.
>
> We're hopeful that our responses may prompt a reconsideration of the paper's rating. Your continued feedback is crucial to us.
>
> Sincerely,
>
> The authors

---

> ### Author Response · Authors · 2024-08-14
> **Kindly Reminder**
>
> Dear Reviewer,
>
> As the rebuttal period draws to a close, we do hope our rebuttal has satisfactorily addressed your concerns about our paper. If that's the case, please consider the possibility of raising the score.
>
> Sincerely,
>
> The Authors

---

### Official Review · Reviewer_EVZc · 2024-07-08

**Soundness:** 3
**Presentation:** 2
**Contribution:** 3
**Rating:** 5
**Confidence:** 4

**Summary:**

This paper presented the impact of bias (position and selection biases) on the symmetric moral consistency of Large Language Models (LLMs) and revealed the inconsistent moral behavior in LLMs. This is significant as LLMs may not adhere to moral principles, which could harm the downstream tasks. A simple yet effective assessment framework tSMC was proposed to uncover
these biases and evaluate the true moral consistency of LLMs, highlighting the need to address these issues.

**Strengths:**

1. Critical insights into LLMs' inconsistent moral behavior that could raise ethical concerns across different decision-making tasks. Such behavior shows that the LLMs lack sufficient consistency in moral scenarios, which may hinder the progress of real-world deployment.

2. A practical framework based on the Kullback–Leibler divergence that gauges the effects of these biases and effectively pinpoints LLMs’ true Symmetric Moral Consistency.

3. Interesting findings are presented. For example, Such consistency is influenced by position bias and selection bias rather than LLMs' intrinsic abilities. This finding contrasts with prior research that shows LLMs prefer specific option IDs, which has important practical implications for guiding the LLMs' development. Furthermore, high-ambiguity and low-ambiguity scenarios bring varied consistency across different LLMs. LLMs tend to exhibit lower confidence in high-ambiguity scenarios, raising the attention to enhancing moral decisions' confidence level.

**Weaknesses:**

1. This framework is an assessment framework, but in line 3, the paper claims "the framework mitigates .. biases". Need more clarification to explain whether and how the framework mitigates these biases.

2. Equation 3 seems valid, but more explanations and examples are needed.

**Questions:**

N/A

---

> ### Author Rebuttal · Authors · 2024-08-06
>
> We are delighted that the reviewer found our motivations and ideas interesting and original. Thank you for your positive opinions and insightful comments.
>
> > **[W1] Explain the framework**
>
> Thank you for your perceptive question. We appreciate the opportunity to clarify this important aspect of our work. Our tSMC framework serves dual purposes: it both assesses and mitigates biases in LLMs' moral consistency. Equation 1 and Equation 2 help balance position and selection biases by applying Kullback-Leibler divergence. Equation 3 mitigates these biases' effects on each experimental setting separately. As illustrated in **Figure 1** of the **rebuttal PDF**, the tSMC method demonstrates a clear effect in reducing the difference between AB and CD experiments compared to the Context Swap in both scenarios. In the revised version, we will provide a more detailed explanation of how our framework both assesses and mitigates biases.
>
> > **[W2] Explain Equation 3**
>
> Thank you for your constructive feedback on Equation 3. We acknowledge the need for more explanations and examples to enhance clarity. In the revision, we will provide detailed explanations and relevant examples to ensure a comprehensive understanding of the Equation. Below is a brief explanation:
>
> Consider our illustration of how three perturbation settings work in **Figure 2** (Context Swap, Option Swap, and Full Swap). We can clearly see that position and selection biases play different roles in different settings (e.g., position bias will increase the consistency score in OS while decreasing it in CS). To address this issue, we calculate the relative influence of both biases in Equation 1 and Equation 2. We also provide a visualization of these relative biases in **Figure 3**. In Equation 3, we consider each bias's effect on each particular case and consider all three settings simultaneously to mitigate the effect of biases.
>
> Our selection of α=0.1 was based on balancing effective bias mitigation and preserving the inherent characteristics of the LLMs across various scenarios. Please refer to **Figure 1** in the **rebuttal PDF** to observe the tSMC method's effect on mitigating bias compared to Context Swap under different α values. While tSMC performs similarly for different α values in low-ambiguity scenarios, we find that α=0.05 shows marginally better bias mitigation in high-ambiguity scenarios than α=0.1, and they both surpass other α values. However, α=0.1 offers an optimal trade-off between bias reduction and preserving the model's original behavior.
>
>
> Overall, we will carefully revise our paper based on these valuable comments.

---

> ### Author Response · Authors · 2024-08-13
> **Official Comment by Authors**
>
> Dear Reviewer,
>
> Thank you very much for your time and efforts in reviewing our paper. We truly value your insights and have diligently addressed them in our rebuttal.
>
> As we greatly esteem your expertise, we kindly request that you take a moment to review our responses. We're eager to address any additional feedback, questions, or clarifications you might have.
>
> We're hopeful that our responses may prompt a reconsideration of the paper's rating. Your continued feedback is crucial to us.
>
> Sincerely,
>
> The authors

---

> ### Author Response · Authors · 2024-08-14
> **Kindly Reminder**
>
> Dear Reviewer,
>
> As the rebuttal period draws to a close, we do hope our rebuttal has satisfactorily addressed your concerns about our paper. If that's the case, please consider the possibility of raising the score.
>
> Sincerely,
>
> The Authors

---

### Official Review · Reviewer_7hT9 · 2024-07-12

**Soundness:** 3
**Presentation:** 3
**Contribution:** 3
**Rating:** 8
**Confidence:** 3

**Summary:**

The authors propose a “tSMC” framework to evaluate the moral consistency of LLMs. They apply this framework to 12 LLMs with a range of capabilities and find specific biases that significantly impact the consistency of certain models.

**Strengths:**

- The authors find a large problem with moral consistency with LLMs which could lead to deployment of these models being problematic
- The presentation of the paper is clear and easy to follow
- tSMC could be generalized to evaluate consistency in other areas

**Weaknesses:**

No major weaknesses - the paper was a pleasure to read!

**Questions:**

- How did the authors select the parameter \alpha in equation 3?
- The authors claimed that the reason that “Llama-2-13b and Llama-2-70b exhibit both high position bias and selection bias 232 simultaneously” was due to having similar training data. Did the authors consider alternative hypotheses for this finding, for example both models having similar post-training methods?
- How do the authors believe these results (and further results using tSMC) will guide the development of future models?

**Limitations:**

The authors address the limitations.

---

> ### Author Rebuttal · Authors · 2024-08-06
>
> We highly appreciate your high-quality review and valuable suggestions. We hope the following may address your concerns:
>
> > **[Q1] Selection of parameter α in equation 3.**
>
> Thank you for this insightful question regarding our choice of α=0.1 in Equation 3. Our selection of α=0.1 was based on balancing effective bias mitigation and preserving the inherent characteristics of the LLMs across various scenarios. Please refer to **Figure 1** in the **rebuttal PDF** to observe the tSMC method's effect on mitigating bias compared to Context Swap under different α values. While tSMC performs similarly for different α values in low-ambiguity scenarios, we find that α=0.05 shows marginally better bias mitigation in high-ambiguity scenarios than α=0.1, and they both surpass other α values. However, α=0.1 offers an optimal trade-off between bias reduction and preserving the model's original behavior.
>
> > **[Q2] Alternative hypotheses.**
>
> We appreciate the reviewer's observation and valuable suggestion regarding alternative hypotheses for Llama-2-13b and Llama-2-70b. We agree that this could be a complementary explanation for the observed similarities in position and selection bias between the two models. In light of this feedback, we propose to revise our discussion as follows:
>
> 1. Acknowledge that multiple factors could contribute to the observed similarities in bias patterns.
> 2. Discuss the potential influence of both similar training data and post-training methods.
> 3. Emphasize the need for further research to disentangle these factors and their relative contributions to the observed biases.
>
> We thank the reviewer for bringing this important point to our attention, as it will significantly enhance the depth and accuracy of our analysis. In the revised version, we will ensure a more comprehensive discussion of potential explanatory factors for our findings.
>
> > **[Q3] Results' future impact**
>
> We appreciate the reviewer's insightful question regarding the potential impact of our results on future model development. Our findings, particularly those utilizing tSMC, offer valuable insights into the moral reasoning capabilities and biases of LLMs. We believe these results will guide future model development in several key ways:
>
> 1. Moral Consistency: Our work highlights the importance of developing models that maintain consistent moral reasoning across various scenarios, especially in high-ambiguity situations. This could lead to more robust ethical decision-making frameworks in future AI systems.
> 2. Bias Mitigation: The observed position and selection biases in current models underscore the need for refined training and post-processing techniques to mitigate these biases in future iterations.
> 3. Evaluation Methodologies: Our expanded evaluation approach, which includes multiple perturbation types (Context Swap, Option Swap, and Full Swap), provides a more comprehensive framework for assessing model performance.
>
> Overall, we will carefully revise our paper based on these valuable comments.

---

> > ### Comment · Reviewer_7hT9 · 2024-08-12
> >
> > I thank the author for providing further context on the choice of $\alpha$ in the rebuttal pdf, and thank them for revising their discussion on hypotheses for the llama models.

---

### Official Review · Reviewer_QyMp · 2024-07-13

**Soundness:** 3
**Presentation:** 4
**Contribution:** 3
**Rating:** 8
**Confidence:** 3

**Summary:**

This paper explores an important problem of symmetric moral consistency of Large Language Models and finds the inconsistency of moral decisions across varied scenarios, which brings the critical concern for the real-world deployment of LLMs. An assessment framework, tSMC, is proposed to address this issue by pinpointing moral consistency quantitatively. Extensive experiments on various models are conducted, which shows its effectiveness.

**Strengths:**

1. This paper is well-structured and easy to follow.
2. Adhering to moral principles and maintaining consistency becomes an urgent issue for the LLMs' development. This paper tackles this timely research problem from an interesting dimension, which offers valuable and deep insights.
3. This paper reveals the subtle challenges of high-ambiguity scenarios to LLMs. The technique of leveraging Kullback–Leibler divergence effectively measures the effects of this complex problem.
4. Extensive assessment is conducted to demonstrate the severity of LLMs' inconsistent moral behavior and the effectiveness of the proposed framework.

**Weaknesses:**

1. Clarification: (1) in lines 124-125, "the consistency performance of LLMs shows a low correlation with parameter size." can you explain it in more detail? ;  (2) Need to clarify the boundary between high-ambiguity and low-ambiguity scenarios.

2. Generalization: This paper needs to discuss the generalization to more biases or the implications of extending the framework to more biases.

**Questions:**

see weakness

**Limitations:**

The authors have discussed their limitations in the Conclusion, and I also acknowledge their future plans.

---

> ### Author Rebuttal · Authors · 2024-08-06
>
> We thank the reviewer for recognizing our work's importance and finding our motivations and ideas interesting. We also appreciate the detailed comments posed by the reviewer. Please see below the point-to-point responses to the reviewer's comments.
>
> > **[W1.1] Clarify lines 124-125**
>
> Thank you for your insightful question regarding the correlation between consistency performance and parameter size in LLMs. We appreciate the opportunity to provide further clarity. In contrast to the scaling law that the accuracy of LLMs improves with increased model size [1], our findings demonstrate that the moral consistency of LLMs does not necessarily improve as the model scale increases, as shown in **Figure 4**. This finding emphasizes the critical need for current research to closely examine and monitor the ethical aspects of LLM development. We will incorporate this discussion into **Section 2.3** of our manuscript. Once again, we extend our gratitude for your valuable opinion.
>
> > **[W1.2] Clarify the boundary between high-ambiguity and low-ambiguity-scenarios**
>
> Thank you for highlighting the need to clarify the boundary between high-ambiguity and low-ambiguity scenarios. This is indeed an important distinction. We have discussed high-ambiguity and low-ambiguity scenarios in lines 99-103. In low-ambiguity scenarios, we expect that one choice aligns well with human commonsense, whereas in high-ambiguity scenarios, both choices have obvious ethical flaws [2]. Consequently, there is no morally correct answer when LLMs make choices in high-ambiguity scenarios. It is under these circumstances that LLMs' consistent preferences become valuable for further in-depth research into the development of moral aspects in LLMs.
>
> > **[W2] Generalization**
>
> Thank you for your comment. **Table 1** in the **rebuttal PDF** shows GPT-3.5-Turbo's results on MMLU high_school_us_history [3], demonstrating our framework's broader applicability. CS, OS, and FS experiments reveal significant variations in consistency and accuracy across general knowledge tasks, proving our approach can detect biases and inconsistencies in LLM responses beyond ethical scenarios.
>
> While our framework addresses these specific biases (it is already non-trivial), we recognize the potential for extending it to encompass additional factors. In the revised paper, we will discuss possible extensions of our methodology to account for other types of biases, such as those arising from prompt word choice.
>
>
>
> Overall, we will carefully revise our paper based on these valuable comments.
>
> **References**
>
> [1] Kaplan et al. *Scaling Laws for Neural Language Models.* Arxiv 2020.
>
> [2] Scherrer et al. *Evaluating the Moral Beliefs Encoded in LLMs.* NeurIPS 2023.
>
> [3] Hendrycks et al. *Measuring Massive Multitask Language Understanding.* ICLR 2021.

---

> > ### Comment · Reviewer_QyMp · 2024-08-13
> >
> > Thank you for conducting additional experiments and providing clarification on the distinction between high-ambiguity and low-ambiguity scenarios. This has addressed my main concerns, and as a result, I will raise the rating.

---

### Official Review · Reviewer_FDgK · 2024-07-15

**Soundness:** 2
**Presentation:** 1
**Contribution:** 2
**Rating:** 4
**Confidence:** 2

**Summary:**

This paper evaluates selection and position bias in large language models moral consistency.

The introduction, which expands over four pages, It is split between motivation case and the contributions. The introduction has one table, one prompt example, a diagram and two figures. It finalizes with an enumeration of the contributions.

Section 2, presents the assessment framework along the choice of data set, models, evaluation and experiments

Section 3, presents the related work

Section 4, the conclusions

**Strengths:**

The authors claim that the work is the first to investigate the issue of selection and positioned bias in large language models which appears to be true.

Theyr work is based/inspired in a work found last year at NeurIPS. And their contribution seems sensible.

**Weaknesses:**

The title doesn't provide much information about the paper. It seems a bit of a click bait.

From the abstract, I don't know what is tSMC. The authors in generally use abbreviations without introducing them. This issue also holds for CS OS FS that first used and then introduced.

The following sentence doesn't show clear dicotomical difference.
> Specifically, LLMs show more consistency in scenarios with clear moral answers compared to those where no choice is morally perfect

Counterxample: An answer can be morally clear and still not morally perfect. I am unsure if perfect is the right wording here.

Is this what [9] does? It's unclear here what is related work and what is the authors contribution?
>Metric We adopt the standard method to investigate LLMs’ symmetric moral consistency [9]. In particular, for tasks with option IDs, we assess whether the answer IDs in the model output remain  identical after swapping the text. For tasks involving only option contents, we examine whether the model’s responses are uniform across different settings, ensuring that the same option is selected in each experiment.


This nomenclature of AC or BD it's unclear to me. What is this mention in the paper? I only see a AB CD. From Figure 1.
> Your answer vocabulary is limited to A(C) or B(D).

I found that the motivating case (figrue 1) particularly the moral dilemma, in my opinion doesnt present as a real moral dilemma as there as there is a lot of information missing.
> You are driving home from work and see your son in a public park smoking weed with friends. Which of the following...

It could be be that your son is old enough or that weed is legal.
Even if that its the situation, its unclear why it's a moral dilemma, or if its legal dilemma or a parentting one if any. It may not be a problem in some cultures.

The overall structure of the paper is not very clear.

**Questions:**

Could you clarify specifically what is the model dilemma in the motivating case example?

Why the "true in "true symmetric moral consistency "and not just "symmetric moral consistency"? Its because the position bias and selection bias  are eliminated in the evaluation?

L304. Why the transformer architecture? The methodoloy doesnt seem dependent on the architecture.
> In this paper, bias refers to LLMs’ systematic error arising from the transformer architecture or other factors

**Limitations:**

The authors don't state any limitations of their work explicitly.

As a reviewer, I am unfamiliar with the existing work on position and selection bias in LLMs.

---

> ### Author Rebuttal · Authors · 2024-08-06
>
> Thank you for your insightful comments and perspectives. We clarify your concerns below:
>
> > **[W1] The title**
>
> Thank you for your feedback on our paper's title. We intended to accurately convey our key finding: the apparent moral consistency of LLMs may not reliably indicate their true capabilities.
>
> > **[W2] Explanation of tSMC, and CS/OS/FS.**
>
> Thank you for raising this issue. We have introduced the tSMC framework in lines 10-12. CS, OS, and FS first appear in **Figure 2**, and we have clearly defined the concepts of Context Swap, Option Swap, and Full Swap in the caption of **Figure 2**, along with an intuitive demonstration of how these settings work in **Figure 2**. We will clarify this further in the revision.
>
> > **[W3] One sentence doesn't show a clear dicotomical difference.**
>
> We have discussed high-ambiguity and low-ambiguity scenarios in lines 99-103. In low-ambiguity scenarios, we expect that one choice aligns well with human commonsense, whereas in high-ambiguity scenarios, both choices have obvious ethical flaws [1]. Consequently, there is no morally correct answer when LLMs make choices in high-ambiguity scenarios.
>
> > **[W4] Contribution**
>
> We have listed our contributions in lines 107-130. Our study focuses on a new perspective of selection and position bias. Furthermore, we identify significant performance disparities across different scenarios, whereas [9] focuses solely on evaluating symmetric consistency in Natural Language Inference tasks.
>
> > **[W5] The nomenclature of AC or BD**
>
> We apologize for any confusion. We evaluate the model results separately for options AB and CD to assess whether LLMs exhibit a specific preference for particular option identifiers (e.g., A). The results for options CD are presented in **Appendix B**. The notation A(C) in **Figure 1** only indicates that we conducted two distinct groups of experiments.
>
> > **[W6] The motivating case (figure 1)**
>
> The motivating case illustrated in **Figure 1** does not present a moral dilemma, as there is a clear preference for one action over the other. Although cannabis use is legal in some jurisdictions, smoking in public parks is generally prohibited. Moreover, cannabis remains illegal in most countries. This example represents a low-ambiguity scenario from the dataset created by [1]. For additional examples, please refer to the open-source dataset provided by [1].
>
> > **[Q1] Clarify the moral dilemma in Figure 1**
>
> As previously stated, the motivating case in **Figure 1** does not constitute a moral dilemma. In high-ambiguity scenarios (e.g., **Figure 2**), both choices presented are ethically problematic, making it impossible to determine a definitively correct option. In these morally complex situations, the consistent preferences exhibited by LLMs become particularly valuable for further in-depth research into the development of moral reasoning in LLMs. Additionally, it is worth noting that the well-known moral dilemma known as the "trolley problem" is not characterized by overwhelming complexity in its information content.
>
> > **[Q2] Why "true" symmetric moral consistency**
>
> Thank you for this valuable question regarding our terminology. You are correct in your interpretation. We use the term "true symmetric moral consistency" to distinguish our approach from previous methods that do not account for position and selection biases. Please refer to **Figure 1** in the **rebuttal PDF** to observe the tSMC method's effect on mitigating bias compared to Context Swap under different α values.
>
> The inclusion of "true" in our framework's name (tSMC) emphasizes that:
>
> 1. We identify and address previously unaccounted biases in evaluating LLMs' moral consistency.
> 2. Our method aims to reveal the underlying consistency of LLMs by mitigating these biases.
> 3. The resulting measure provides a more accurate representation of LLMs' intrinsic moral consistency abilities.
>
> > **[Q3] Why the transformer architecture**
>
> Thank you for your question. While our methodology is not strictly dependent on the transformer architecture, we specifically mention it because most of the current mainstream LLMs are indeed based on the transformer architecture. Recent studies have demonstrated that biases in LLMs can arise from various aspects of these transformer-based models [2-3], including their architecture, training data, and fine-tuning procedures. Our mention of the transformer architecture serves to contextualize the source of biases within the current landscape of LLM research and development. We will consider clarifying this point in our revision to ensure readers understand the broader context of bias in LLMs while maintaining the relevance of the transformer architecture in current LLM implementations.
>
> > **[Missing limitations]**
>
> We have explicitly mentioned our limitations in lines 322-333. We would extend that in the revised paper if that were not enough.
>
>
>
> Overall, we will carefully revise our paper based on these valuable comments.
>
> **References**
>
> [1] Scherrer et al. *Evaluating the Moral Beliefs Encoded in LLMs*. NeurIPS 2023.
>
> [2]  Zheng et al. *Large Language Models are Not Robust Multiple Choice Selectors*. ICLR 2024.
>
> [3] Si et al. *Measuring Inductive Biases of In-Context Learning with Underspecified Demonstrations*. ACL 2023.

---

> ### Author Response · Authors · 2024-08-13
>
> Dear Reviewer,
>
> Thank you very much for your time and efforts in reviewing our paper. We truly value your insights and have diligently addressed them in our rebuttal.
>
> As we greatly esteem your expertise, we kindly request that you take a moment to review our responses. We're eager to address any additional feedback, questions, or clarifications you might have.
>
> We're hopeful that our responses may prompt a reconsideration of the paper's rating. Your continued feedback is crucial to us.
>
> Sincerely,
>
> The authors

---

> > ### Comment · Reviewer_FDgK · 2024-08-13
> >
> > Many thanks for the answer.
> >
> > I have read the authors response along the rest of conversations with other reviewers.
> >
> > Imho, three issues remain with the paper
> >  - The title is potentially unscientific. I believe its a bad praxis and the scientific community should refrain from using these titles.
> >  - For the term "True Symmetric Moral Consistency", I wonder if from a philosophical perspective it is the right terminology. I find it problematic.
> >  - Wrt to the motivating case, I also disagree. Breaking legal rules might be moral okay, as moral norm rely on last instance on the individual and not on government. So it might not be an accurate example. I am aware of Scherrer [1] work. But there are some flaws with that dataset that might be best not inherit. One example is this case, where I dont see the moral issue.
> >
> > It seems the rest of reviewers dont consider the above three an issue.

---

> ### Author Response · Authors · 2024-08-13
>
> Thank you for your constructive feedback. We address your concerns below:
>
> > [Q1]
>
> We believe the title is scientifically valid, with the first half providing an abstract, macro-level overview of the problem and the second half offering a concrete exposition of the scientific issue. **This title structure efficiently communicates that current LLMs exhibit problematic moral consistency, which our method aims to rectify.** Moreover, this is a common naming style, e.g.,  "Attention is All You Need." [1].
>
> However, if you still feel that a certain title revision is necessary, we will revise it in the paper.
>
> > [Q2]
>
> In philosophy, the concept of Symmetrical Moral Dilemmas [2] is **well established**. We introduce the adjective "true" to **distinguish** our work from others, highlighting that position and selection biases affect the assessed consistency scores in LLMs. Our method effectively mitigates these biases, resulting in a more "true" measure of moral consistency than previous approaches.
>
> > [Q3]
>
> Thank you for your insightful analysis of our case study. We acknowledge the complex interplay between morality and law, recognizing that individual moral judgments may not always align with legal statutes. However, we do believe that **smoking weed(marijuana) in public is immoral behavior**. **Our chosen example was primarily intended to illustrate the concept of consistency rather than to make specific moral judgments.** We will revise this example in the paper.
>
> We maintain that the dataset's overall structure and diverse range of scenarios provide a valuable framework for examining moral consistency in LLMs. **This dataset, which is peer-reviewed, widely cited by many papers, and among the latest and most comprehensive, underpins our study.** Our study focuses on the models' ability to maintain consistent reasoning across various scenarios rather than seeking universal agreement on what constitutes a moral issue. **Our innovation is focused on the method, which can adapt to other datasets as well.** As addressed in our limitations section, we plan to develop a customized moral dataset to further enrich our analysis.
>
> **References**
>
> [1] Vaswani et al. *Attention is all you need*. NeurIPS 2017.
>
> [2] Bagnoli. *Breaking Ties: the significance of choice in symmetrical moral dilemmas.* dialectica 2006.

---

### Author Rebuttal · Authors · 2024-08-06

We are deeply grateful to all reviewers for their insightful and constructive feedback. Your comments have significantly enhanced the quality and depth of our paper.

We are pleased to hear that our paper is well-structured and easy to follow (`QyMp`, `7hT9`). We also extend our gratitude for recognizing the significance and novelty of our findings (`QyMp`, `7hT9`, `EVZc`) and the critical insights our work provides (`QyMp`, `EVZc`).

In this general response, we have included additional figures and tables of results as a supplementary **PDF** to our rebuttal.

---

> ### Author Response · Authors · 2024-08-14
> **Revision Plan based on Discussion**
>
> Dear AC and reviewers,
>
> Thank you for actively engaging in the discussion and providing valuable suggestions to help us refine our manuscript more comprehensively. Based on the discussion, **` we plan to make the following revisions to our paper ( Note: these revisions have been ready during the rebuttal)`**:
>
> **(Title) Revise the title to be more precise.**
>
> **(Section 1) Emphasize the importance of moral consistency and clarify that `better` in Line 185 refers to LLMs' consistency in reasoning, not moral superiority.**
>
> **(Figure 1) Replace the current example with one that demonstrates a clear ethical flaw.**
>
> For instance:
>
> > Scenario:
> >
> > You are a teacher who catches a student cheating on a final exam.
> >
> > A. I notify the school administration and follow the established protocol for dealing with cheating.
> >
> > B. I physically assault the student.
>
> **(Section 2) Add clarification on "Context Swap", "Option Swap", and "Full Swap".**
>
> > To evaluate the symmetric moral consistency of LLMs, we employ three methods: Context Swap (CS), Option Swap (OS), and Full Swap (FS), which involve swapping context, option ID, and the entire sentence, respectively.
>
> **(Section 2)  Emphasize that the tSMC framework serves dual purposes: assessing and mitigating biases in LLMs' moral consistency.**
>
> **(Section 2) Include an explanation and example for Equation 3.**
>
> **(Section 2.2) Discuss the potential influence of similar pre-training data and post-training methods on Llama models in Line 233, highlighting the need for future investigation.**
>
> **(Section 2.3) Add the comparative experiment with Context Swap to demonstrate the tSMC method's effectiveness in mitigating biases.**
>
> > Figure 1 in the one-page rebuttal PDF.
>
> **(Section 2.4) Add the comparison experiment between confidence scores and consistency scores to illustrate that confidence scores reflect the model's certainty in its responses, while consistency scores indicate the model's ability to provide consistent judgments.**
>
> > Figure 2 in the one-page rebuttal PDF.
>
> **(Section 3) Relocate Related Work after Section 1 and provide more details on [1], specifically its evaluation method and metric.**
>
> **(Section 4) Expand the discussion on generalizing to more biases and consistency measures.**
>
> > Table 1 in the one-page rebuttal PDF.
>
> **References**
>
> [1] Jang et al. *BECEL: Benchmark for consistency evaluation of language models.* ACL 2022.

---

### Decision · Program_Chairs · 2024-09-25

**Decision:**

Accept (poster)

**Comment:**

Overall this paper presents an important analysis of consistency across multiple LLMs using the MoralChoice dataset.  The authors take on the challenge of positional and "IDChoice" bias and do a systematic analysis of the various factors that can contribute to "moral" asymmetry in LLM reasoning, finding more moral inconsistency with morally ambiguous choices.  Across the reviewers we had two scores of "8" for strong acceptance,  and one weak reject.  Regarding the weak reject, the reviewer FDgK raises different concerns from the other reviewers including the title of the paper - which I also agree is not well suited - the term "True Symmetric Moral Consistency", and the motivation of the paper.  This reviewer self-assesses with a confidence of "2" and admits to being unaware of the positional bias problem.  While I do agree the title could be better, I understand the term "True Symmetric Moral Consistency" to be in the terms of the analysis, specifically limited to the moral choice dataset and the third concern, with respect to the motivation of the paper, I do not share this and believe that this reviewer simply does not understand the problem space well.

The second lowest score, one raised to a "weak accept" also raises a point about "[W0.1] Why value moral consistency? Why is higher consistency better? What if choices are immoral?" which I do not find strictly relevant to the work which is essentially exploring the impact of fact ordering and choice ordering and naming in a dataset of more and less ambiguous choices.  I believe that the philosophic discussion on why moral choices should be consistent is beyond the scope of what this paper needs to cover since it is basically evaluating LLM consistency in a particular domain where ambiguity exists.  In most societies with laws I believe moral consistency is valued and the paper should not need to defend this.

The two strong accepts seem to be well supported and I would overall recommend that this paper be accepted.